# Lipoic Acid Combined with Melatonin Mitigates Oxidative Stress and Promotes Root Formation and Growth in Salt-Stressed Canola Seedlings (*Brassica napus* L.)

**DOI:** 10.3390/molecules26113147

**Published:** 2021-05-25

**Authors:** Hafiz Muhammad Rashad Javeed, Mazhar Ali, Milan Skalicky, Fahim Nawaz, Rafi Qamar, Atique ur Rehman, Maooz Faheem, Muhammad Mubeen, Muhammad Mohsin Iqbal, Muhammad Habib ur Rahman, Pavla Vachova, Marian Brestic, Alaa Baazeem, Ayman EL Sabagh

**Affiliations:** 1Department of Environmental Sciences, COMSATS University Islamabad, Vehari Campus, Vehari 61100, Pakistan; Rashadjaveed@cuivehari.edu.pk (H.M.R.J.); mazharali@cuivehari.edu.pk (M.A.); mzfaheem@hotmail.com (M.F.); Muhammadmubeen@cuivehari.edu.pk (M.M.); mohsiniqbal6369@gmail.com (M.M.I.); 2Department of Botany and Plant Physiology, Faculty of Agrobiology, Food and Natural Resources, Czech University of Life Sciences Prague, 16500 Prague, Czech Republic; skalicky@af.czu.cz (M.S.); vachovap@af.czu.cz (P.V.); 3Department of Agronomy, Muhammad Nawaz Shareef University of Agriculture, Multan 60000, Pakistan; fahim.nawaz@mnsuam.edu.pk; 4Department of Agronomy, College of Agriculture, University of Sargodha, Sargodha 40100, Pakistan; drrafi1573@gmail.com; 5Department of Agronomy, Bahauddin Zakariya University, Multan 60000, Pakistan; atiqjugg@gmail.com; 6Crop Science Group, Institute of Crop Science and Resource Conservation (INRES), University Bonn, 53113 Bonn, Germany; mhabibur@uni-bonn.de; 7Department of Plant Physiology, Slovak University of Agriculture, 94901 Nitra, Slovakia; marian.brestic@uniag.sk; 8Department of Biology, College of Science, Taif University, P.O. Box 11099, Taif 21944, Saudi Arabia; aabaazeem@tu.edu.sa; 9Department of Agronomy, Faculty of Agriculture, University of Kafrelsheikh, Kafr el-Sheikh 33516, Egypt

**Keywords:** adaptation, oxidative stress, lipoic acid, melatonin, antioxidant activities

## Abstract

Lipoic acid (LA) and melatonin (MT) are pleiotropic molecules participating in plant stress resistance by modulating cellular biochemical changes, ion homeostasis, and antioxidant enzyme activities. However, the combined role of these two molecules in counteracting the detrimental impacts of salinity stress is still unknown. In the present study, we determined the effects of exogenous LA (0.5 µM), MT (1 µM) and their combination (LA + MT) on growth performance and biomass accumulation, photosynthetic pigments, enzymatic and non-enzymatic antioxidant activities, and ions homeostatic in canola (*Brassica napus* L.) seedlings under salinity stress (0, 100 mM) for 40 days. The results indicate that exogenous application of LA + MT improved the phenotypic growth (by 25 to 45%), root thickness (by 68%), number of later lateral roots (by 52%), root viability (by 44%), and root length (by 50%) under salinity stress. Moreover, total soluble protein, chlorophyll pigments, the concentration of superoxide dismutase (SOD), catalase peroxidase (CAT), and ascorbic peroxidase (ASA) increased with the presence of salt concentration into the growth media and then decreased with the addition of LA + MT to saline solution. Leaf protein contents and the degradation of photosynthetic pigments were lower when LA + MT treatments were added into NaCl media. The proline and phenol contents decreased in the exogenous application of LA + MT treatments more than individual LA or MT treatments under the salinity stress. The incorporation of LA or MT or a combination of LA + MT to saline solution decreased salinity-induced malondialdehyde and electrolyte leakage. In conclusion, the alteration of metabolic pathways, redox modulation, and ions homeostasis in plant tissues by the combined LA and MT application are helpful towards the adaptation of *Brassica napus* L. seedlings in a saline environment. The results of this study provide, for the first time, conclusive evidence about the protective role of exogenous LA + MT in canola seedlings under salinity stress.

## 1. Introduction

Salinity stress is one of the significant abiotic stresses for crop production, limiting the agricultural crop productivity both in arid and semi-arid regions of the world [1,2]. The rapid increase in ions accumulation in the upper soil layers is due to saline irrigation water, less precipitation than evaporation, global warming, and dry soil [3]. Around 5% of the total arable land is under high salt concentrations. A high exogenous saline environment imposes osmotic stress, ion toxicity, nutritional imbalances [4,5], the production of reactive oxygen species (ROS) [6], cell membrane leakage [7], and biochemical distortion [8], resulting in the inhibition of growth and yield [9]. Therefore, improvement in the plant antioxidant enzymes such as superoxide dismutase, peroxidase dismutase, catalase and ascorbate peroxidase; antioxidant compounds such as ascorbate (ASA), glutathione (GSH), salicylate, α-tocopherol [1,8,10]; and photosynthetic systems such as chlorophyll pigments and electron transport systems [5] may result in normal growth and development under a saline environment. Due to the number of past studies, crop breeding and environmental improvement approaches aiming to induce salt stress tolerance in plants [3,11,12,13] have received considerable attention by the researchers. Thus, different ROS scavengers and antioxidant compounds can enhance the salinity stress tolerance in plants, lowering oxidative stress [2,14].

Lipoic acid (LA) is a non-enzymatic low-molecular-weight antioxidant molecule that can eliminate ROS and regulate the ROS detoxifying systems in plant cells [15]. LA has special properties among all non-enzymatic antioxidants and can work both in oxidized LA and reduced dihydrolipoic acid (DHLA) forms [16]. Due to its solubility in both lipids and water, it regenerates or improves the antioxidant activity of the cytoplasm, and ultimately regulates the antioxidant network in the cell [16]. Although a small amount of LA was detected in the roots and leaves of some plants, e.g., wheat, tomato, potato, and asparagus [17,18,19,20], this amount is not sufficient to rehabilitate the cell metabolites. So, exogenous LA may help combat the detrimental effects of salt stress. Similarly, melatonin (N-acetyl-5-methoxytryptamine) (MT) is a very potent and inducible endogenous hormone, present in both plants and animals, and acts as a modulator against the number of plant stresses [21,22,23,24]. Plants rich in MT concentrations had a more total antioxidative capacity and lower plant-damaging products [25]. In addition to direct protection from ROS, exogenous application of MT also reduced the activities of endogenous antioxidants, including superoxide anion dismutase (SOD), peroxidases (POD), catalase (CAT), ascorbate peroxidase (APX), mono-dehydro-ascorbate reductase (MDHAR), malondialdehyde (MDA), and glutathione reductase [25,26,27,28]. MT also improved the nutrition ingredients of the plant body in stress conditions [29]. Moreover, MT can also protect the plants from salinity stress through a foliar exogenous application or systemic application to plants [30]. Therefore, it is involved in stress-affected plant developmental actions, including germination, flowering, pollination, fruiting, and senescence [31]. MT is also considered an effective bio-stimulator and growth-promoting chemical that can accelerate the seed germination [32], coleoptiles of young seedlings, and growth of roots [33].

Although a lot of work has been carried out on the application of lipoic acid and melatonin against the plant salinity and drought stress, blend application of LA and MT has not yet been studied to understand their synergistic effects in enhancing the enzymatic and non-enzymatic antioxidants in plant cells. So, this study was conducted to apprehend the hypothesis that blend application of lipoic acid and melatonin enhances the antioxidant activities faster than their individual application to saline media and that co-application (LA + MT) regulates the ion homeostasis and improves the root growth under salinity stress.

## 2. Results

### 2.1. Growth Performance, Root Characteristics, and Biomass of Canola Plant

Plant growth performance (plant height, leaf length, and root characteristics) in saline condition for 30-days resulted in visible symptoms in the form of poor plant height and week leaf growth (Figure 1). However, application of LA or MT in saline solution significantly reduced the toxic effects of salinity but the combined application of LA + MT was found to be more effective than the control or individual one. Regarding the shoot dry weight (SDW) and root dry weight (RDW) of canola plants, SDW and RDW were significantly improved (*p* ≤ 0.001 for both) by the addition of LA, MT or LA + MT along with salt treatments (Appendix A and Figure 1C,D). The SDW and RDW were reduced by 34 and 25% in NaCl treatments compared to control. Furthermore, the addition of LA + MT in salt solution increased the SRW and RDW by 44 and 36%, respectively, in comparison to the salt-treated plants. As far as the root growth was concerned (Figure 1 and Figure 2 and Appendix A), the greatest root thickness was observed in the plants where the combined LA + MT was added to the saline environments while the minimum was recorded in the control plants where no treatments were applied (Figure 2E). The maximum number of first lateral roots (Figure 2B) were recorded in the control plants while the salt stress reduced the number of lateral roots by 53% as compared to control plants. However, the combined application of LA + MT significantly increased (by 44%) the number of first lateral roots as compared to plant under salinity stress. The greatest root vitality was noted when combined LA + MT was added to the saline media, which differed significantly from the saline media and control media (Figure 2C). Moreover, under the salt stress, the root elongation was inhibited and reduced by almost 50% as compared to the control, while the addition of combined LA + MT scientifically increased the root length by 44% more than the roots without any treatments (Figure 2D). The interaction of salt and LA, MT or LA + MT showed a non-significant effect (*p* > 0.05) among growth, biomass and root characteristics.

### 2.2. Photosynthetic Pigments, Electrolytic Leakage, and Relative Water Contents

Salt treatments markedly decreased the photosynthetic pigments (chl a, chl b and car contents) in canola leaves by 39, 48, and 44%, respectively, as compared to control, whereas the application of LA, MT or a combination of both (LA + MT) enhanced the photosynthetic pigments by 14, 18 and 25% (chl a), and by 11, 16 and 21% (chl b), respectively, to NaCl-treated plant leaves (Figure 3A). Similarly, car contents were increased by 16, 18, and 27% when LA, MT, and LA + MT were applied, respectively (Figure 3B). Under the salt stress conditions, as shown in Figure 3C, prominent damage to the cell membrane was observed in terms of electrolyte leakage (EL). The application of NaCl to plants notably increased the LE by 63% compared to control, while significantly lower EL contents were found in response to the application of LA (by 31%) or MT (by 25% or LA + MT (by 34%) along with NaCl solution over the individual salt treatments (Figure 3C). Moreover, the exogenous application of LA + MT to saline solution significantly improved the relative water contents by 20% as compared to salinity stress. The maximum relative water contents were recorded in the leaves of control plants, which were statically on par with those of LA + MT treatments (Figure 3D).

### 2.3. Minerals Concentration in Leaves and Root Tissues

Significantly increased (*p* ≤ 0.001 for both ions) Na^+^ and Cl^−^ concentration was noted in the leaves (by 2.5 and 2.2-folds, respectively) and roots (by 3.8 and 4.3-folds, respectively) of canola seedlings under salinity stress to control seedlings (Figure 4A–D). Similarly, a decreased trend (*p* < 0.01 for both ions) was noted in the case of K^+^ and Ca^2+^ as the NaCl was applied compared to control (Figure 4E–H). The addition of LA or MT or LA + MT to salt solution significantly reduced the accumulation of Na^+^ and Cl^−^ ions, both in the leaves and roots (Figure 4A–D); on the other hand, they also increased the concertation of K^+^ and Ca^2+^ in the salt-stressed plants (Figure 4E–H). However, the combined application of LA + MT reduced the concentration of Na^+^ (by 2.1- and 1.9-fold in the leaves and root tissues, respectively) and Cl^−^ more effectively (by almost one fold in the leaves and root tissues) as compared to seedlings in the saline medium. Furthermore, the addition of LA to NaCl solution increased the concentration of K^+^ and Ca^2+^ in leaves, whereas the LA + MT promoted the K^+^ and Ca^2+^ levels in the roots compared to NaCl alone (Figure 4E–H).

### 2.4. Enzymatic and Non-Enzymatic Antioxidant Activities

The data represented in the graphs showed that the application of LA, MT or LA + MT significantly affected the activities of SOD, CAT, POD, and APX in the leaves and root tissues of canola seedlings (Figure 5A–H). The data in Figure 5A depicts that the blend application of LA + MT increased the SOD activity levels (by 47%) around NaCl stress. Moreover, the application of LA to saline solution induced higher SOD activity in the leaf tissues as compared to MT alone (Figure 5A). The addition of antioxidant agents to NaCl solution improved the SOD activity level in the canola seedling roots (Figure 5B) but the more prominent SOD activity was recorded in root tissues treated with MT (by 78%), which was followed by LA + MT (by 72%) and LA (67%) compared to controls (Figure 5B). As far as CAT activity was concerned, the CAT activity level was considerably increased by 34% (in the case of leaf tissues) and 28% (in case of root tissues) as compared to canola seedlings exposed to NaCl alone. Moreover, the treatments with MT produced a significant increase by 18% and 21% in SOD levels in the leaf and root tissues subjected to salt stress (Figure 5C,D).

It was noted that the application of LA + MT to saline solution significantly increased the POD activity in leaf tissues (by 36%) in response to NaCl stress (Figure 5E), but alone, NaCl stress only significantly affected POD activity in the root tissues of canola seedlings (Figure 5F). Moreover, in response to NaCl stress alone, leaf POD activity was 21% higher compared to the control (Figure 5E). Contrary to the activity of POD in the leaf tissues, canola roots exposed to NaCl alone increased 18% POD activity, while the individual application of LA or MT reduced the POD activity by 29 and 25%, respectively, compared to NaCl (Figure 5F). Data presented in Figure 5G illustrates a significant increase of 11% APX activity in the leaf tissues subjected to NaCl stress compared to control. More importantly, the addition of MT and LA + MT to saline medium increased leaf APX activity by 24% and 32%, respectively, compared to plants grown with NaCl alone (Figure 5G). Furthermore, APX activity was found to be 16% higher in the roots of canola seedlings exposed to salt stress compared to the control. Under salinity stress, the application of MT and LA + MT increased the SOD activity by 32% and 26%, respectively, in the roots (Figure 5G).

### 2.5. Proline contents, Phenols, Total Protein Contents, and Malondialdehyde (MDA)

Under salt stress, proline contents were increased by 35% compared to the control treatments, whereas the application of LA + MT alleviated the adverse effects of NaCl by 41% in the leaf tissues of canola seedlings (Figure 6A). In contrast, there was no significant difference of proline concentration in roots among all the applied treatments (Figure 6B); however, proline was increased by 31% in salt-treated roots. The presence of LA alone in the saline solution decreased the proline concentration by 18% with that of salt-stressed plant roots (Figure 6B). On the other hand, an almost similar data trend was noted for proline contents, in the case of canola leaves, with no significant difference in total phenols contents (Figure 6C). Moreover, in salt-treated canola roots, the addition of LA + MT to saline water decreased the phenols by 25% with that of salt-treated plants (Figure 6C). The total phenols increased by 28% in salt-treated canola roots compared to control treatments, whereas the presence of LA + MT decreased phenols concentration by 39% in canola roots subjected to salt stress (Figure 6D). The total protein contents were lower (by 42%) in leaf tissues as compared to the control, while its contents were increased (by 26%) with the supplemental addition of combined LA + MT to NaCl solution (Figure 6E). As regards roots subjected to salt stress, the application of antioxidant agents to saline medium had statistically similar effects on the total protein contents with that of the leaf tissues but maximum improvement (by 18%) in protein contents was noted in the combined supply of LA + MT to NaCl solution. Additionally, it was noted that salinity caused a significant decline in total protein contents (by 30%) in root tissues compared to control treatments (Figure 6F). Furthermore, the application of salt to nutrient solution decreased the endogenous accumulation of MDA in the leaves by 42% than that of leaves of control seedlings, whereas the exogenous supply of combined chemicals (LA + MT) resulted in 26% higher MDA contents to NaCl stressed seedlings (Figure 6G). As far as root tissues were concerned, NaCl-treated roots had lesser MDA contents (by 35%) than that of salt-free plant roots while the external application of both LA + MT increased MDA by 18% compared to NaCl alone treatments (Figure 6H).

## 3. Discussion

Canola plants are considered a salt-sensitive crop; therefore, its growth performance and biomass are severely restricted by salinity (Appendix A, Figure 1, Figure 2 and Figure 3). In the present study, salt stress treatments induced the toxic symptoms on plant height, leaf length (dehydration and necrosis at the leaf edges, and some were twisted or curled), shoot dry weight (SDW), and root dry weight (DRW) of canola plants (Figure 2A–D, Appendix A). However, the deleterious damages of NaCl stress were alleviated (no visible symptoms on the leaves with significant improvement of leaf turgor and osmotic pressure (Appendix A) by the blend application of LA + MT into saline solution [8,34,35]. The higher SDW and RDW might be due to higher shoot fresh weight and root fresh weight in the blend treatments of LA + MT to a saline solution (Appendix A and Figure 2). Moreover, data presented in Figure 2 shows that the inhibitory effect of salinity stress on root thickness, lateral roots, root viability, and root length was alleviated by the combined addition of LA + MT to saline media [35]. Our findings are inconsistent with those of Huang, Chen, Zhao, Ding, Liao, Hu, Zhou, Zhang, Yuan and Yuan [28] and Liang, et al. [36]. They stated that melatonin improved the root growth and development under a saline environment. Moreover, the root viability under salinity stress was significantly improved in the individual application of MT or blend of LA + MT than LA alone and salinity treatments (Figure 2C). Based on our results, we concluded that a blend application of LA + MT may help strengthen salinity stressed canola roots.

The results of current studies showed that the addition of LA + MT to salt solution inhibited the chlorophyll (Chl a, Chl b, and Car) degradation caused by salinity stress, and delayed the leaf senesce by the improvement of nutrients (Appendix A) and the leaf turgor pressure and osmotic pressure (Appendix A). Farhangi-Abriz and Ghassemi-Golezani [3] and Zoufan, Azad, Rahnama Ghahfarokhie and Kolahi [9] demonstrated that chlorophyll loss or degradation can occur prematurely, resulting in the leaf senescence if initiated by external factors. This study exhibited that the photosynthetic pigments were remarkably reduced under salt stress (Figure 3A–C). On the other hand, the NaCl damaged the plant leaf cell membrane, showing a higher amount of electrolyte leakages (EL) but the exogenous application of LA or MT or their combination improved the cellular structure of the cell membrane under salinity stress by decreasing the EL. Our results gave the complementary evidence that the combined application of LA + MT has a synergistic role in improving, maintaining, or scavenging hydroxyl ions, superoxide ions, and singlet oxygen towards the seedling phenology, photosynthetic pigments [28,36], and EL showed the ability of canola seedlings to withstand in a saline environment [37].

Ion chemistry under salinity stress is one the most vital phenomena for proper plant functioning and other metabolic activities. Salinity stress is known as the toxic accumulation of Na^+^ and Cl^−^ ions in the tissues of plants’ leaves and roots (Figure 4). Overall data trend indicated that all the exogenously applied biomolecules significantly modulated the ion homeostasis [38] in the plant tissues, but it was noted that LA + MT proved itself as a strong free radical scavenger [6] rather than MT or LA alone, and also improved the plant growth, i.e., SFW and RFW (Appendix A) [5]. Presumably, the combined application of LA + MT reduced the accumulation of Na^+^ and Cl^−^ in the leaf and root tissues of canola plants [3] than the application of LA or MT alone. On the other hand, LA + MT also improved the K^+^ and Ca^+^ in the plant tissues [3]. Moreover, this considerable K^+^ concentration in the cytosol may improve the membrane stability [6], and, hence, lower the EL (Figure 4B). The results of this study concluded about the novel role of MT indirect absorption of K^+^ by canola plants under a saline environment.

Under a salinity stress environment, the exogenous addition of LA + MT improved and enhanced the activities of SOD, CAT, and POD in the leaf and root tissues by inhibiting the intercellular accumulation of H_2_O_2_ and ultimately acted as direct ROS scavengers (Figure 5A–F). These results conclude that combined application of lipoic acid and melatonin could modulate intracellular ROS concentration by maintaining a steady state of ROS levels, and may have a protective role, sustaining the cell membrane integrity against salinity stress. Our findings are in line with the studies conducted in Campos, Oliveira, Pereira and Farnese [14], soya bean seedlings [3], and pak choi (*Brassica Chinensis* L.) [6]. Moreover, APX can directly scavenge the O_2_^−1^ from plant cells and, hence, reduce the H_2_O_2_ level. Our study results illustrated that LA + MT treatments to canola seedlings increased the APX level both in leaf and root tissues under salinity stress (Figure 5G,H). So, the higher level of APX may help the soya bean seedlings to alleviate oxidative stress damage under NaCl stress [3]. The results of this present study exhibited that the addition of LA + MT plays a vital role in enzymatic and non-enzymatic antioxidants systems in protecting from oxidative damages under salinity stress conditions by enhancing their activities or levels.

Total phenols (TPh), proline, and total protein (TP) have a vital role in the osmotic adjustment of plant tissues during stress conditions. Our study results showed that the application of LA + MT to canola seedlings considerably decreased these osmoprotectants compared with salt-stressed or LA- or MT-treated canola seedlings (Figure 5A–F). However, an increase in these osmolytes was more prominent in leaves than root tissues. Meanwhile, LA + MT also modulated the oxidative stress but was not very effective in comparison to MT-applied plant tissues [28]. Similar results were noted by Carlson, et al. [39], who stated that a higher concentration of organic solutes recycled the free ions and radicals in the plant cell membrane. However, the synergetic effect of LA + MT was also noted during this study. In the present study, the exogenous application of LA + MT to a saline environment significantly decreased the production of MDA (Figure 6G,H) in both the leaves and root tissues of canola seedlings. Moreover, the excess ROS damages caused by salinity stress resulted in a higher concentration of MDA contents (Figure 6G).

## 4. Materials and Methods

### 4.1. Biological Material, Experimental Design, and Salinity Treatments

Seeds of *Brassica napus* L. cultivar (cv. MS007, marketed by FMC Pvt. Ltd., Vehari Pakistan) were taken from the local grain market of Vehari, Pakistan. *Brassica napus* L. (commonly known as Canola) seeds were treated with 5% sodium hypochlorite solution for 10 min before the start of the experiment to remove any kind of foreign materials, then washed with distilled water five times and spread in a lab for air-drying. After this, these seeds were ready for sowing. A hundred seeds were sown in Petri-plates (10 seeds per Petri plate), placing filter papers below and above the seeds. The filter paper was moistened with half-strength Hoagland solution (HS) with the supplemental addition of 0 or 100 mM NaCl or melatonin (µM) (MT) or lipoic acid (LA) treatments (0.5 µM). Germination was completed within 15 to 18 days. The canola seedlings were allowed to grow until the shoot attained 5 to 6 cm height and root attained 7 to 8 cm. These seedlings were now ready for transplanting. Before transplanting the seedlings in plastic containers, the containers (v. 7 L) were filled with half-strength Hoagland solution [40] and the following treatments: (1) control, only HS (CT); (2) HS + 100 mM NaCl (HS + S); (3) HS + 100 mM NaCl + 0.5 mM lipoic acid (S + LA); (4) HS + 100 mM NaCl + 1µM melatonin (S + MT); (5) HS + 100 mM NaCl + 0.5 mM lipoic acid + 1 µM melatonin (S + LA + MT). The concentrations of lipoic acid (Sigma Aldrich, St. Louis, MO, USA) and melatonin (Sigma Aldrich, St. Louis, MO, USA) used in this experiment were based on preliminary experiments carried out at the Department of Environmental Sciences, COMSATS University Islamabad Vehari-Campus, Vehari Punjab, Pakistan (data not given). The salt stress was imposed after seven days of plant transplanting (plant hardening) to Hoagland solution. In case of any death of plant/s, the plant/s was replaced from the stock. The experiment was arranged using the completely randomized design (CRD) with four replications (four plants per replicate of each cultivar). The seedlings in the Hoagland solution were aerated with small pumps (aquarium pumps) until the experiment was harvested. The Hoagland solution was replaced after every 10 days and the experiment lasted for 30 days (almost four weeks). The canola seedlings were placed in the greenhouse at 70 to 85% humidity with a 09/15 day/night cycle. The plants were harvested and subjected to the study of different parameters such as plant growth and root characteristics, photosynthetic pigments, membrane stability, proline contents, antioxidants, and ion concentrations.

### 4.2. Growth and Root Characteristics Measurement

The growth characteristics, i.e., plant height, leaf length, and root characteristics (i.e., root length shoot length, root thickness, and root vitality) were measured with the help of ImageJ software by following the methods of Rasband [41] and Tajima and Kato [42]. This software converts the pixels into mm. The root vitality was recorded using the triphenyltetrazolium chloride (TTC) method [43].

### 4.3. Photosynthetic Pigments

Photosynthetic pigments (chlorophyll a (chl a), chlorophyll b (chl b), chl a + chl b, and carotenoid (car) contents) were measured by following the method as described by Palta [44]. Leaf pigment concentrations were quantified by using the formula [45] as described in Equations (1)–(3);
chlorophyll a (mg/g FW) = (11.75 × A663 − 2.35 × A645) × 50/500(1)
chlorophyll b (mg/g FW) = (18.61 × A645 − 3.96 × A663) × 50/500(2)
carotenoid (mg/g FW) = ((1000 × A470) − (2.27 × Chl a) − (81.4 × Chl b)/227) × 50/500(3)
FW = fresh weight of leaf, A 663, A645, A470 = absorbance wavelength on a spectrophotometer

### 4.4. Determination of Relative Membrane Permeability and Proline Contents

Relative membrane permeability was measured at crop harvesting by following Yang, et al. [46] The electrolyte leakage in terms of percentage was quantified by following the formula EC% = (C1/C2) × 100. The proline content analysis in the plant leaves was measured using the method of Bates, et al. [47].

### 4.5. Extraction and Assays of SOD, CAT, POD, APX, and MDA

Approximately 0.5 g of fresh plant tissues were crushed using a mortar and pestle (precooled) in 2 mL 0.1 M potassium phosphate buffer (pH 7 with 0.1 mM EDTA) and centrifuged at 10,000× *g* for 20 min at 4 °C. The supernatant was stored on ice and used to determine different antioxidant activities.

The activity of superoxide dismutase (SOD; EC 1. 15.1.1) exposed its presence by inhibiting the photochemical reduction of nitro blue tetrazolium (NBT) and was determined according to Gupta, et al. [48] This mixture was placed under a light source (15 W, fluorescent lamps) at 78 μmol m^−2^ s^−1^ for 15 min. The absorbance of the irradiated solution was recorded at 560 nm using a spectrophotometer (Lambda 25 Perkin Elmer Singapore). Catalase (CAT, EC 1.11.1.6) and peroxidase dismutase (POD, EC 1.11.1.7) activities were determined by following the methods as described by Aebi and Lester [49] and Panda, et al. [50].

Ascorbate peroxidase (APX, EC 1.11.1.11) activity was recorded by measuring the decrease in absorbance of ascorbic acid at 290 nm continuously for 90 s. The change in absorbance was due to oxidation of ascorbate in the reaction (extinction coefficient of 2.8 mM^−1^ cm^−1^), and calculation of APX activity was assayed using Amako, et al.’s method [51]. The lipid peroxidation was recorded in terms of malondialdehyde (MDA) contents, and readings were taken at 532 and 600 nm. The MDA concentrations were calculated by following the Equation (4) developed by Heath and Packer [52]:MDA (µmol g^−1^ FW) = ((A532 − A600)/155,000) × 106(4)

### 4.6. Determination of Total Protein Contents and Total Phenolic Contents

Total protein contents were determined by following the Bradford procedure [53]. We used the bovine serum albumin as the standard regent. Total phenolic contents were determined in the plant extracts using the Folin–Ciocalteu reagent [54]. Briefly, plant extract (50 mg) was added into a falcon tube containing Folin–Ciocalteu reagent (0.5 mL) and distilled water (7.5 mL) was added. The aliquot was incubated at room temperature for 10 min. The mixture was heated at 40 °C for 20 min after adding the 20% Na_2_ CO_3_ (1.5 mL). The absorbance was recorded at 755 nm and expressed in terms of gallic acid equivalent (mg/100 g of plant extract).

### 4.7. Determination of Mineral Concentration in Roots and Leaves

The collected samples of plant tissues (roots and leaves) were washed with distilled water and then placed in hot air at 70 ± 2 °C, until a constant weight was achieved. The oven-dried ground plant samples (0.1 g) were digested by acid-based digestion (sulfuric acid and hydrogen peroxide mixture, 2 mL) and the minerals (Na^+^, K^+^, and Ca^2+^) were determined by following the methods of Wolf [55]. The ions in the digested material were determined with the help of BWB-XP flame photometer (BWB Technologies, Newbury, UK). For Cl^−^ analysis, 1.0 g of the ground shoot and root samples were extracted in 10 mL of distilled water at 80 °C for 4 h. Cl^−^ concentration was determined with a chloride analyzer (Corning, 925, Columbus, OH, USA).

### 4.8. Statistical Analyses

Recorded data were analyzed using the statistical program STATISTIX (10.1.V, Analytical Software 2105 Miller Landing Rd, Tallahassee, FL 32312, USA) and the analysis variance of the techniques was measured by the Fisher analysis of variance techniques [56]. The individual means of the treatments were compared by using the least significant difference test at 5% probability level. The error bars on the columns were the means of four replicates (each replicate contained four plants).

## 5. Conclusions

In conclusion, the alteration of metabolic pathways, redox modulation, and ions homeostasis in plant tissues by the combined LA and MT application are helpful towards the adaptation of *Brassica napus* L. seedlings in a saline environment. The results of this study provide, for the first time, conclusive evidence about the protective role of exogenous LA + MT in canola seedlings under salinity stress. These results will facilitate plant breeders to produce genetically engineered plants that may produce a sufficient amount of those molecules.

## Figures and Tables

**Figure 1 molecules-26-03147-f001:**
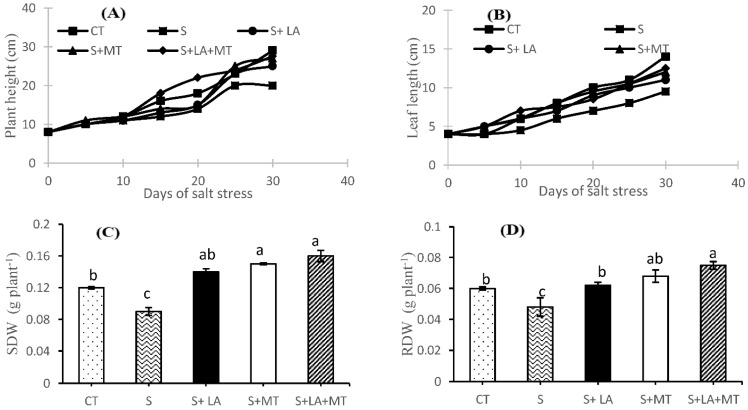
The effect of lipoic acid (LA), melatonin (MT), and their combination (LA + MT) under salt stress on plant height (**A**), leaf length (**B**), shoot dry weight (SDW) (**C**), root dry weight (RDW) (**D**) of *Brassica napus* L. seedlings exposed to salinity (100 mM NaCl) for 30 days. Abbreviation; CT, control; S, salinity; S + LA, salinity + lipoic acid; S + MT, salinity + melatonin; S + LA + MT, salinity + lipoic acid + melatonin. Error bars on each column show the ± SE of three replication samples. Different letters on each bar indicate a significant difference according to the LSD test (*p* ≤ 0.05).

**Figure 2 molecules-26-03147-f002:**
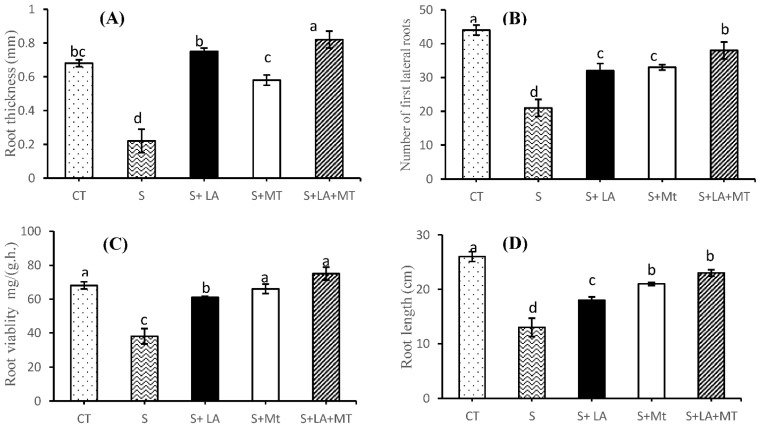
The effect of lipoic acid (LA), melatonin (MT), and their combination (LA + MT) under salt stress on root thickness (**A**), number of first lateral roots (**B**), root viability (**C**), root length (**D**) of *Brassica napus* L. seedling exposed to salinity (100 mM NaCl) for 30 days. Abbreviations are given in the Figure 1. Error bars on each column show the ± SE of three replication samples. Different letters on each bar indicate a significant difference according to the LSD test (*p* ≤ 0.05).

**Figure 3 molecules-26-03147-f003:**
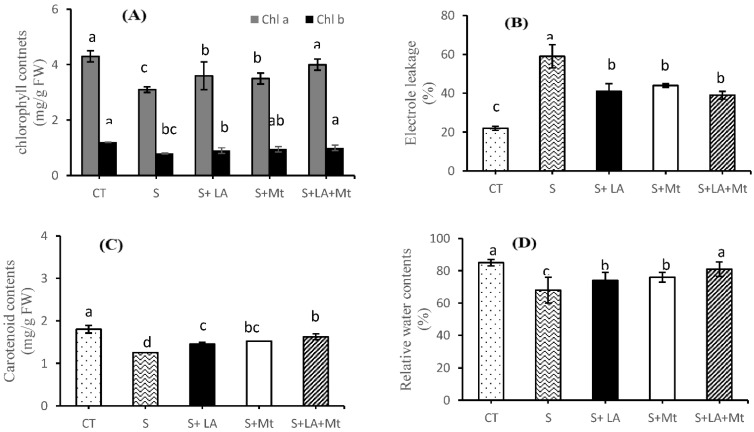
The effect of lipoic acid (LA), melatonin (MT), and their combination (LA + MT) under salt stress on chlorophyll contents (**A**), electrolyte leakage (**B**), carotenoid contents (**C**), relative water contents (**D**) of *Brassica napus* L. seedlings exposed to salinity (100 mM NaCl) for 30 days. Abbreviations are given in the Figure 1. Error bars on each column show the ± SE of three replication samples. Different letters on each bar indicate a significant difference according to the LSD test (*p* ≤ 0.05).

**Figure 4 molecules-26-03147-f004:**
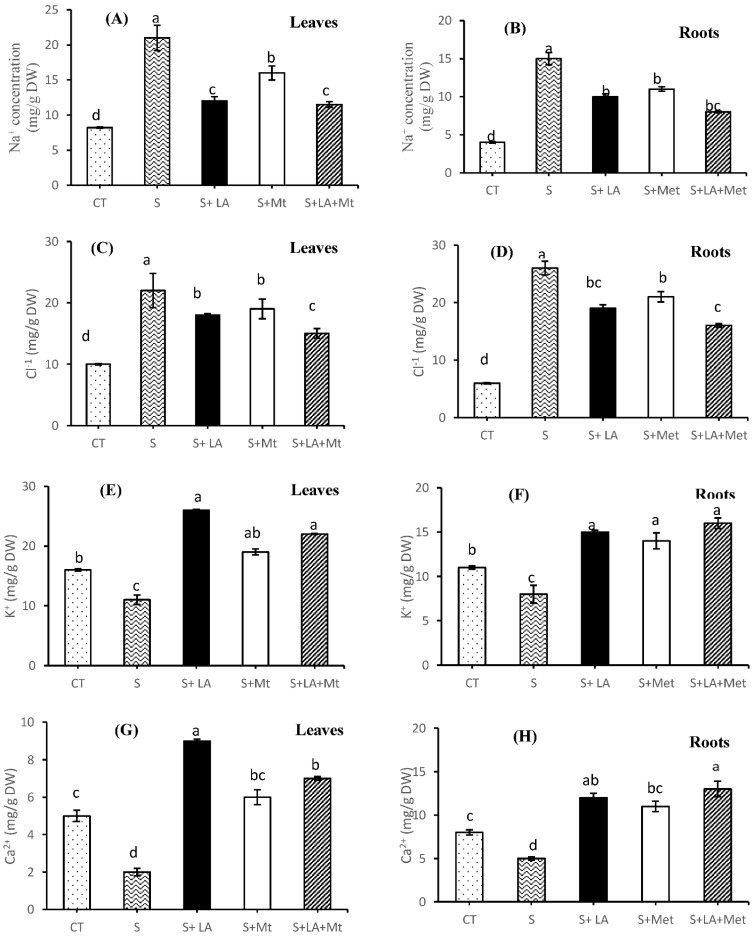
The effect of lipoic acid (LA), melatonin (MT), and their combination (LA + MT) under salt stress on Na^+^ concentration in leaves (**A**) and roots (**B**), Cl^−1^ concentration in leaves (**C**) and roots (**D**), K^+^ concentration in leaves (**E**) and roots (**F**) and Ca^+2^ concentration in leaves (**G**) and roots (**H**) of *Brassica napus* L. seedlings exposed to salinity (100 mM NaCl) for 30 days. Abbreviations are given in the Figure 1. Error bars on each column show the ± SE of three replication samples. Different letters on each bar indicate a significant difference according to LSD test (*p* ≤ 0.05).

**Figure 5 molecules-26-03147-f005:**
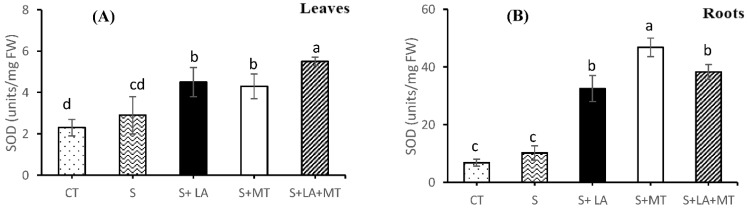
The effect of lipoic acid (LA), melatonin (MT) and their combination (LA + MT) under salt stress on SOD in leaves (**A**) and roots (**B**), CAT in leaves (**C**) and roots (**D**), POD in leaves (**E**) and roots (**F**) and APX in leaves (**G**) and roots (**H**) of *Brassica napus* L. seedlings exposed to salinity (100 mM NaCl) for 30 days. Abbreviations are given in Figure 1. Error bars on each column show the ± SE of three replication samples. Different letters on each bar indicate a significant difference according to LSD test (*p* ≤ 0.05).

**Figure 6 molecules-26-03147-f006:**
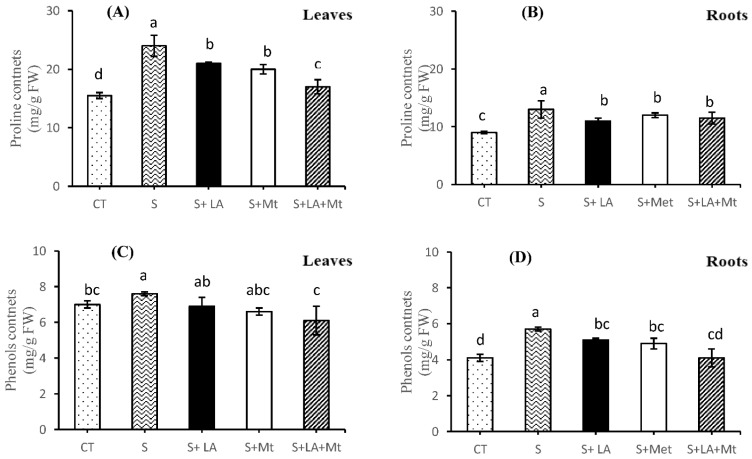
The effect of lipoic acid (LA), melatonin (MT) and their combination (LA + MT) under salt stress on proline contents in leaves (**A**) and roots (**B**), total phenol contents in leaves (**C**) and roots (**D**), total protein contents in leaves (**E**) and roots (**F**) and malondialdehyde (MDA) contents in leaves (**G**) and roots (**H**) of *Brassica napus* L. seedlings exposed to salinity (100 mM NaCl) for 30 days. Abbreviations are given in the Figure 1. Error bars on each column show the ± SE of three replication samples. Different letters on each bar indicate a significant difference according to LSD test (*p* ≤ 0.05).

## Data Availability

Not applicable.

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
