# Peer review of "Lipoic Acid Combined with Melatonin Mitigates Oxidative Stress and Promotes Root Formation and Growth in Salt-Stressed Canola Seedlings (Brassica napus L.)"

_molecules, 2021, doi:10.3390/molecules26113147_

Round 1

Reviewer 1 Report

Manuscript by Javeed H.M.R. et al describing the influence of lipoic acid and melatonin on the oxidative stress parameters and plant growth in salt -stressed Canola seedlings in interesting and it deserves to be published i Molecules however after some minor changes.

Abstract part of the manuscript is well constructed and it contains all the necessary information.

Introduction part of the manuscript is well written and it also contains essential information, but authors could add some information regarding oxidative stress in alininty stress in canola species - just a few sentences.

Results are well presented and described.

Discussion - line 267 there is "blind application of LA+MT" - should it be "blend application of LA+MT"?

Materials and methods:

line 335 - there is I µM, should it be 1µM?

Could authors explain in more details why did they choose such LA and MT concentrations?what was tested concentration range?

I couldn't find methodology for TPh - total phenols and TP total proteins content, where is it?

Authors could also estimate protein oxidation as a marker of protein damage, for example thiol group content. If it is not possible now, they could find information in the literature and discuss how LA and MT could possibly influence on oxidative damage of proteins. The same with ROS. If it is not possible now to estimate ROS content - authors should add some information in Discussion part of the manuscript.

Author Response

Reviewer-1 Comments and their answer

The Authors wish to extend their special thanks to the reviewers for their valuable comments which will surely improve the quality of manuscript. All the comments were incorporated and addressed as given below;

Abstract

No comments

Introduction

No comments

Results

No comments

Discussion

C-1: line 267 there is "blind application of LA+MT" - should it be "blend application of LA+MT"?

R-1: Incorporated (Line 291, Page10)

Materials and Methods

C-1: line 335 - there is I µM, should it be 1µM?

R-1: corrected (Line 258, Page 12)

C-2: Could authors explain in more details why did they choose such LA and MT concentrations? what was tested concentration range?

R-2: The cementation of LA and MT was calibrated in the laboratory in a petri plate experiment before the experiment with different treatment. Then the best treatment on the basis seedlings growth and development is further used for hydroponic experiment.

C-3: I couldn't find methodology for TPh - total phenols and TP total proteins content, where is it?

R-3: Added (Line 422, Page13)

Conclusion

No comments

Reviewer 2 Report

The topic is interesting and the results are promising. However:

  • Not enough replicates were used in the experiments – should be at least 5.
  • Context is missing. Why was this salt concentration chosen? Why were these particular lipoic acid and melatonin concentrations chosen? Furthermore, the authors write “Our findings are inconsistent with those of Huang, Chen, Zhao, Ding, Liao, Hu, Zhou, Zhang, Yuan, and Yuan [32] and Liang, et al. [42]. They stated that the melatonin and lipoic acid improved the root growth and development under a saline environment.” – but I cannot find any mention of lipoic acid in either reference (to be precise, I can access the full text of the first reference, but only the abstract of the second paper because the rest is in Chinese). Why was lipoic acid included in this analysis?
  • Effects of addition of lipoic acid and melatonin in absence of salt stress were not examined, but they are important, particularly in order to understand the effect of lipoic acid on ion uptake.
  • Melatonin is already known to increase osmotic stress tolerance, but to the best of my knowledge, the effect of exogenous addition of lipoic acid to plants was not tested before. Lipoic acid is a co-factor that should affect mitochondrial metabolism. And if it is added exogenously – at 0.5 µM, i.e., in a concentration that would be suited for exogenous addition of phytohormones, but would not seem to support a direct effect as antioxidant –, it has a _dramatic_ effect on the uptake of potassium and calcium ions (Figure 4)? This result really has to be discussed in detail. Results on rats (https://www.ncbi.nlm.nih.gov/pmc/articles/PMC5336776/) show an effect of lipoic acid on ATP-dependent potassium channels, but I have not found anything for plants.
  • The discussion should end with a paragraph summarizing the conclusions.

Minor comments:

Where did you obtain your lipoic acid? Was it a mix of enantiomers, or did you obtain a purified enantiomer?

Figures 1-5: The abbreviations ”CT” and ”S” have to be explained in the legends. (Yes, I can guess what they mean, but I shouldn’t have to guess).

Figure 1A, B: difference between the “CT” and the “S” label is too weak. Use open boxes for one of them. And we need the standard deviations!

Figure 4E: there must be a mistake. The standard deviations are minimal, but there is no significant difference between S+LA and S+LA+MT? Hard to believe.

Line 27 et al.: “Brassica napus” should be always in italics

Line 28: What is “phenotypic growth” as opposed to “growth”?

Line 30: “ under salinity stress.” – provide salt concentration here

Line 32: “of salt concentration into the growth media” – should be “…in the growth media”

Lines 33-35: “ Besides, lower leaf protein contents and lower degradation of photosynthetic pigments were observed in LA+MT treatments than those in NaCl treatments.” – Please specify: Do you mean that leaf protein contents and degradation of photosynthetic pigments was lower when LA and MT were added to NaCl-containing media? What happened when LA and MT were added in the absence of NaCl?

Lines 35-36: “The proline and phenol contents decreased in the exogenous application of LA+MT treatments than LA or MT alone treatments.” – Is this with salt or without? And the comparison is not clear – did you forget a “more” in the first half of the sentence?

Line 80: “through a foliar exogenous application or systemic application to plants” – what is systemic application? Added to the watering solution? Are melatonin and/or lipoic acid transported in the xylem?

Line 97ff: “However, application of LA or MT in saline solution significantly reduced the toxic effects of salinity but the combined application of LA+MT” – Context is missing. First, which salt concentration? (Yes, it is in the figure legend, but it should have been already in the abstract). Second, how were the LA and MT concentrations chosen? Third, what is the effect of LA and MT on growth in absence of salt stress?

Lines 139/140: “Application of NaCl to plants notably increased the LE by 63 % compared...” – EL, not LE

Lines 144/145: “The maximum relative water contents were recorded in the leaves of control plants (Fig. 3D)” – not true; according to figure 3D, there was no significant difference in relative water contents between control plants and salt-stressed plants with LA+MT.

Author Response

Reviewer-2 Comments and their answer

The Authors wish to extend their special thanks to the reviewers for their valuable comments which will surely improve the quality of manuscript. All the comments were incorporated and addressed as given below;

General comments

C-1: Not enough replicates were used in the experiments – should be at least 5.

R-1: We appreciate the comment of the reviewers but the four (4) replications are enough in the hydroponic studies. Many past studies used the three and four replications.

C-2: Context is missing. Why was this salt concentration chosen? Why were these particular lipoic acid and melatonin concentrations chosen? Furthermore, the authors write “Our findings are inconsistent with those of Huang, Chen, Zhao, Ding, Liao, Hu, Zhou, Zhang, Yuan, and Yuan [32] and Liang, et al. [42]. They stated that the melatonin and lipoic acid improved the root growth and development under a saline environment.” – but I cannot find any mention of lipoic acid in either reference (to be precise, I can access the full text of the first reference, but only the abstract of the second paper because the rest is in Chinese). Why was lipoic acid included in this analysis?

R-2: This salt concentration is based on the lab pert plate experiment and this canola cultivar tolerate up to 100 mM NaCl, so this concentration is used. The information related reference was corrected

C-3: Effects of addition of lipoic acid and melatonin in absence of salt stress were not examined, but they are important, particularly in order to understand the effect of lipoic acid on ion uptake.

R-3: We appreciated the suggestion of the reviewer

C-4: Melatonin is already known to increase osmotic stress tolerance, but to the best of my knowledge, the effect of exogenous addition of lipoic acid to plants was not tested before. Lipoic acid is a co-factor that should affect mitochondrial metabolism. And if it is added exogenously – at 0.5 µM, i.e., in a concentration that would be suited for exogenous addition of phytohormones, but would not seem to support a direct effect as antioxidant –, it has a _dramatic_ effect on the uptake of potassium and calcium ions (Figure 4)? This result really has to be discussed in detail. Results on rats (https://www.ncbi.nlm.nih.gov/pmc/articles/PMC5336776/) show an effect of lipoic acid on ATP-dependent potassium channels, but I have not found anything for plants.

R-4: We appreciated the suggestion of the reviewer. The explanation is given in the discussion section

C-5: The discussion should end with a paragraph summarizing the conclusions.

R-5: The separate conclusion is give at the end of manuscript. I think no need of conclusion at the end of discussion.

Minor Comments

Abstract

C-1: Where did you obtain your lipoic acid? Was it a mix of enantiomers, or did you obtain a purified enantiomer?

R-1: The lipoic acid was purchased from the local market (Sigma-Aldrich) in the form of Alpha-lipoic acid (ALA).

C-2: Figures 1-5: The abbreviations ”CT” and ”S” have to be explained in the legends. (Yes, I can guess what they mean, but I shouldn’t have to guess).

R-2: The abbreviations were added at the legend of Fig. 1 and consider for all figures as mentioned.

C-3: Figure 1A, B: difference between the “CT” and the “S” label is too weak. Use open boxes for one of them. And we need the standard deviations!

R-3: Corrected

C-4: Figure 4E: there must be a mistake. The standard deviations are minimal, but there is no significant difference between S+LA and S+LA+MT? Hard to believe.

R-4: I have checked the information again, the figure is corrected. The difference is present but it was a minor difference

C-5: Line 27 et al.: “Brassica napus” should be always in italics

R-5: Corrected

C-6: Line 28: What is “phenotypic growth” as opposed to “growth”?

R-6: Yes

C-7: Line 30: “ under salinity stress.” – provide salt concentration here

R-7: Salt concentrations are added

C-8: Line 32: “of salt concentration into the growth media” – should be “…in the growth media”

R-8: We appreciated the comments of the reviewer

C-9: Lines 33-35: “ Besides, lower leaf protein contents and lower degradation of photosynthetic pigments were observed in LA+MT treatments than those in NaCl treatments.” – Please specify: Do you mean that leaf protein contents and degradation of photosynthetic pigments was lower when LA and MT were added to NaCl-containing media? What happened when LA and MT were added in the absence of NaCl?

R-9: Corrected as seen in the Abstract.

C-10: Lines 35-36: “The proline and phenol contents decreased in the exogenous application of LA+MT treatments than LA or MT alone treatments.” – Is this with salt or without? And the comparison is not clear – did you forget a “more” in the first half of the sentence?

R-10: The word salinity stress added in the text.

Introduction

C-1: Line 80: “through a foliar exogenous application or systemic application to plants” – what is systemic application? Added to the watering solution? Are melatonin and/or lipoic acid transported in the xylem?

R-1: Systemic application mean transported through xylem (Page 2 Line 83)

C-2: Line 97ff: “However, application of LA or MT in saline solution significantly reduced the toxic effects of salinity but the combined application of LA+MT” – Context is missing. First, which salt concentration? (Yes, it is in the figure legend, but it should have been already in the abstract). Second, how were the LA and MT concentrations chosen? Third, what is the effect of LA and MT on growth in absence of salt stress?

R-2: No past study is present about the combine application of LA+MT, that’s why, this study was carried out to explore the combined effect of LA+MT on the plants under salinity stress. No past review of literature about the combine application of LA+MT is present. This present study will provide the base for the future studies about the combine application of LA+MT. Secondly, about how the concentrations of LA+MT was chosen, it was explained in detail in the Materials and Methods Section.

Results

C-1: Lines 139/140: “Application of NaCl to plants notably increased the LE by 63 % compared...” – EL, not LE

R-1: as compared to control (no salinity), Explained in the Results sections (Page 4 Line 148)

C-2: Lines 144/145: “The maximum relative water contents were recorded in the leaves of control plants (Fig. 3D)” – not true; according to figure 3D, there was no significant difference in relative water contents between control plants and salt-stressed plants with LA+MT.

R-2: Yes, you are right, but maximum was in control and statistically similar in LA+ MT. Added and corrected (Page 4 Line 152-153)

Discussion

No Comments

Material and Methods

No Comments

Conclusion

No Comments

All the comments are incorporated successfully. Now, the authors are optimistic about the acceptance of the manuscript for publication in the Molecules Journal.

Round 2

Reviewer 2 Report

Lines 34-35:

“Besides, lower leaf protein contents and lower degradation of photosynthetic pigments were observed in when LA+MT treatments were added into NaCl media.” – remove “treatments”

Lines 36-38:

“The proline and phenol contents decreased in the exogenous application of LA+MT treatments than LA or MT alone treatments under the salinity stress.” – Better: “Unter salinity stress, exogenous application of LA+MT led to stronger decrease of proline and phenol contents than application of LA or MT alone.”

Figure legends 2-6:

“Abbreviations are given in the Fig. 1.” – remove “the”

This manuscript is a resubmission of an earlier submission. The following is a list of the peer review reports and author responses from that submission.